

# A symbol and coaction for higher-loop sunrise integrals

Andreas Forum and Matt von Hippel⋆

Niels Bohr International Academy, Niels Bohr Institute,
University of Copenhagen, Copenhagen, Denmark

⋆ mvonhippel@nbi.ku.dk

## Abstract

We construct a symbol and coaction for the $l$-loop sunrise family of integrals, both for equal-mass and generic-mass cases. These constitute the first concrete examples of symbols and coactions for integrals involving Calabi-Yau threefolds and higher. In order to achieve a symbol of finite length, we recast the differential equations satisfied by these integrals in a unipotent form. We augment the integrals in a natural way by including ratios of maximal cuts $\tau_i$. We discuss the relationship of this construction to constructions of symbols and coactions for multiple polylogarithms and elliptic multiple polylogarithms, and its connection to notions of transcendental weight.

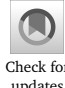

# 1  Introduction

Two mathematical structures, the symbol and coaction on multiple polylogarithms, have revolutionized the way we compute loop-level Feynman diagrams [1–6]. Multiple polylogarithms are iterated integrals over rational functions, and the symbol and coaction are operations that disassemble these iterated integrals, revealing their branch cut and derivative structure and trivializing identities between them. Since their introduction to the physics community in ref. [2], the symbol and coaction have been key tools, whether for simplifying results [5], clarifying new structure [7–24], reformulating differential equations [25], or in special cases bootstrapping complete quantities from minimal constraints [26–38].

As the physics community considers more complicated processes with more loops or more complicated kinematic dependence, we have begun to encounter Feynman diagrams which cannot be expressed in terms of multiple polylogarithms. The oldest such example is a two-loop propagator correction involving an exchange of massive particles [39]. Now known as the two-loop sunrise, this diagram has been the subject of intensive study [40–59]. The two-loop sunrise is part of a broader class of diagrams that evaluate to integrals involving elliptic curves. Recently, a formalism has been proposed for these integrals that parallels that previously developed for multiple polylogarithms, involving iterated integrals over these elliptic curves [60, 61]. In ref. [62], the authors of the latter paper developed a symbol and coaction for these functions. This construction was based on general recipes for building symbols and coactions for motivic periods described in ref. [63], modifying the ideas somewhat and specializing them to the elliptic case.

Higher-loop propagator corrections can involve more complicated functions yet. A particularly well-studied example is a higher-loop analogue of the sunrise integral, also known as a banana integral [64–66]. While the equal-mass case at three loops can be represented in terms of functions from the elliptic toolbox, namely iterated integrals of modular forms, [67, 68], more general three-loop sunrise integrals and sunrise integrals at higher loops require more complicated functions. These integrals have been observed to have a surprising connection to Calabi-Yau manifolds [69–71], with a Calabi-Yau $l-1$-fold appearing at $l$ loops. This connection has recently been exploited to construct differential equations and series expansions for these integrals [72–74]. The latest paper in that series presented higher-loop sunrise integrals in terms of a particularly suggestive iterated integral form, involving elements of a particular basis for their maximal cuts called the Frobenius basis.

Inspired by this result, we investigate the possibility of applying the general construction implied by the treatment in ref. [62] to higher-loop sunrise integrals, making use of the structure of their differential equations as laid out in ref. [74]. A key requirement for the construction of a symbol is the existence of functions that satisfy a unipotent differential equation, that is a homogeneous differential equation formulated with a nilpotent matrix. We find that the formulation of ref. [74] does not immediately satisfy these requirements, but it does once the basis is suitably enlarged. To enlarge the basis, we make a natural choice of adding ratios of maximal cuts $\tau_i$, analogous to the modulus $\tau$ of the torus associated with the elliptic curve at two loops. With this enlarged basis and an appropriate choice of normalization, we find that we can construct a symbol and coaction for the higher-loop sunrise integrals. This represents

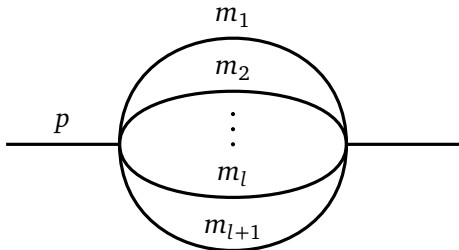

Figure 1: The $l$-loop sunrise integral.

the first specific examples of a symbol and coaction constructed for integrals involving Calabi-Yau manifolds with dimension greater than one. We find that our construction has an unusual relationship to the notions of "transcendental weight" in the literature, which are largely inspired by polylogarithmic Feynman integrals. We comment on this relationship, related to different choices one can make in the symbol and coaction construction, and highlight the importance of choices that match particular practical goals.

The paper is organized as follows. We begin in section 2 with some preliminaries: subsection 2.1 defines our integral of interest, subsection 2.2 reviews the general symbol and coaction construction we will be using, and subsection 2.3 describes the form of differential equations that we will employ. Finally, subsection 2.4 presents a basis of integrals with useful properties deriving from their relevance to Calabi-Yau manifolds. We then present our symbol and coaction constructions, first for the equal-mass case in section 3, then for the generic-mass case in section 4. We conclude in section 5, discussing some implications of our work and directions for further study. We also include one ancillary file, `SunriseSupplementaryMaterial.nb`, with MATHEMATICA functions that output the symbol and coaction for the equal-mass sunrise integral through five loops. The ancillary file is accessible via the version of this paper on the arXiv at https://arxiv.org/abs/2209.03922.

## 2 Preliminaries

### 2.1 The $l$-loop Sunrise Integral

The $l$-loop sunrise integral is depicted in Figure 1. It has $l + 1$ propagators, with masses $m_1 \ldots m_{l+1}$. In this paper we consider integrals of this type with general propagator powers,

$$I_{\underline{\nu}}(\underline{x}, D) = \int \prod_{r=1}^{l} \frac{d^D k_r}{(i\pi)^{D/2}} \prod_{j=1}^{l+1} \frac{1}{(q_j^2 - m_j^2)^{\nu_j}}, \tag{1}$$

with $q_j$ the momentum flowing through the relevant propagator. $\underline{\nu}$ denotes the vector of integer $\nu_j$ propagator powers, and $\underline{x}$ expresses the dependence of the integral on momentum invariants:

$$\underline{x} = \{p^2, m_1^2, \ldots m_{l+1}^2\}. \tag{2}$$

In practice, it is always possible to factor out an overall power of one of these momentum invariants, typically chosen to be $p^2$. The nontrivial kinematic dependence of these integrals is then encoded in a vector of dimensionless ratios,

$$\underline{z} = \{m_1^2/p^2, \ldots m_{l+1}^2/p^2\}. \tag{3}$$

The set of $l$-loop sunrise integrals with general propagator powers are related by integration-by-parts identities. They can thus be expressed in a basis of master integrals. In

$D = 2 - 2\epsilon$ dimensions, there will be a total of $2^{l+1} - 1$ independent master integrals. We will represent these in different bases as we continue, but we begin with the basis initially discussed in ref. [74]. This basis consists of $l + 2$ sectors, of which $l + 1$ reduce to tadpole integrals,

$$J_{l,i}(\underline{z}, \epsilon) = -\frac{(-1)^{l+1}}{\Gamma(1 + l\epsilon)} (p^2)^{l\epsilon} \epsilon^l I_{1,\dots,1,0,1,\dots,1}(\underline{x}, 2 - 2\epsilon) = -\frac{\Gamma(1 + \epsilon)}{\Gamma(1 + l\epsilon)} \prod_{j=1, j\neq i}^{l+1} z_j^{-\epsilon}, \qquad (4)$$

and one top sector containing the remaining integrals, indexed by $\underline{k} \in \{0, 1\}^{l+1}$,

$$J_{l,\underline{0}}(\underline{z}, \epsilon) = \frac{(-1)^{l+1}}{\Gamma(1 + l\epsilon)} (p^2)^{1 + l\epsilon} I_{1,\dots 1}(\underline{x}, 2 - 2\epsilon), \qquad (5)$$

$$J_{l,\underline{k}}(\underline{z}, \epsilon) = (1 + 2\epsilon)\dots(1 + k\epsilon) \partial_{\underline{z}}^{\underline{k}} J_{l,\underline{0}}(\underline{z}, \epsilon), \qquad (6)$$

where $k = |\underline{k}|$.

If all of the propagator masses are equal $m_i^2 = m^2$, then there are only $l + 1$ distinct master integrals. We begin by representing them in the following basis,

$$J_{l,0}(z, \epsilon) = \frac{(-1)^{l+1}}{\Gamma(1 + l\epsilon)} (m^2)^{l\epsilon} \epsilon^l I_{1,\dots 1,0}(\underline{x}, 2 - 2\epsilon) = -\frac{\Gamma(1 + \epsilon)}{\Gamma(1 + l\epsilon)}, \qquad (7)$$

$$J_{l,1}(z, \epsilon) = \frac{(-1)^{l+1}}{\Gamma(1 + l\epsilon)} (m^2)^{1 + l\epsilon} I_{1,\dots 1}(\underline{x}, 2 - 2\epsilon), \qquad (8)$$

$$J_{l,k}(z, \epsilon) = (1 + 2\epsilon)\dots(1 + k\epsilon) \partial_z^{k-1} J_{l,1}(z, \epsilon), \qquad (9)$$

where $2 \leq k \leq l$.

The number of independent master integrals also changes for special values of $\epsilon$. In particular, the number of independent master integrals in the top sector are reduced to $2^{l-1} - \binom{l-2}{\lfloor \frac{l+2}{2} \rfloor}$ for $\epsilon = 0$. This reduction is due to the fact that the derivatives in eq. 6 generate the cohomology groups in $H_{\text{hor}}^{l-1-k,k}(M_{l-1}^{CI}), k = 0, \dots, l-1$. The cohomology group $M_{l-1}^{CI}$ contains linear dependencies of these derivatives, which means that the number of independent ones are instead lower than what is found in eq. 6.

## 2.2 Symbols and coactions for motivic periods

In this section we review the construction of symbols and coactions for motivic periods outlined in ref. [62], based on those authors' interpretation of ref. [63].

The symbols we are familiar with from multiple polylogarithms take the following form: for a function of transcendental weight $w$ whose derivative satisfies

$$dF_w = \sum_i F_{w-1}^i d\log\phi_i, \qquad (10)$$

one obtains a symbol

$$S(F_w) = \sum_i S(F_{w-1}^i) \otimes \phi_i. \qquad (11)$$

Symbols of this sort must have a few key features. In order for the definition to be recursive in this way, the functions the symbol is defined for must satisfy a homogeneous system of differential equations, so that derivatives do not map outside the space of functions for which symbols are defined. In order for symbols to be of finite length, the recursion must terminate.

Let us assume we have a basis of $n$ linearly independent functions $\underline{I}$, which we can think of as integrations of a basis of forms $\xi$ over a common contour,

$$\underline{I} = \left( \int_\gamma \xi_1, \dots \int_\gamma \xi_n \right), \tag{12}$$

that satisfy a homogeneous system of differential equations,

$$d\underline{I} = A\underline{I}, \tag{13}$$

where $A$ is an $n \times n$ matrix of one-forms. In order for the symbol to be of finite length, we require that $A$ is a nilpotent matrix, that is that there is some positive integer $k$ such that $A^k = 0$. A differential equation with this property is called unipotent.

Once we have a basis of functions with this property, we can apply the recipe of refs. [62,63] to define a symbol and coaction. This recipe will be unfamiliar to readers used to symbols and coactions for polylogarithms, so before reviewing the method it is worth clarifying a few points. For polylogarithms, we are used to thinking of the symbol as the maximal iteration of a coaction: a coaction applied repeatedly until we have broken down our functions into the smallest possible logarithmic pieces. The entries of a symbol are not actually logarithms, though: logarithms are periods, a pairing of a (rational) form and an integration contour, while symbol entries do not have a contour. This is why we can think of them as defined "up to $i\pi$", or why "the symbol kills constants": a symbol entry does not specify the branch of the logarithm.

Such symbol entries are quite intuitive to manipulate when they resemble logarithms, but when they are more general more care must be taken. This is why the recipe of refs. [62,63] will involve two ingredients. The first, a motivic coaction, breaks up periods into pairs of a period and a pairing of differential forms called a de Rham period. The second ingredient is the symbol, and its role is now to make sense of these de Rham periods, transforming them into formal descriptions of iterated integrals with no specified contour (the so-called "bar construction"). We will now review how to construct each of these ingredients.

The first ingredient, the motivic coaction, takes the following schematic form:

$$\Delta^{\mathrm{m}}([\gamma, \omega]) = \sum_i [\gamma, \omega_i] \otimes [\omega_i, \omega]. \tag{14}$$

The pair $[\gamma, \omega]$ is the notation of ref. [62] for a motivic period, which pairs an integration cycle $\gamma$ with a form $\omega$ on which it can be integrated. $[\omega_i, \omega]$ is a pairing between differential forms that represents a de Rham period.

To define our second ingredient, the symbol map, we begin with the matrix $A$ appearing in our differential equation. From this we define a matrix $T_A$, with entries in the bar construction: formal specifications of iterated integrals.

$$T_A = 1 + [A]^R + [A|A]^R + [A|A|A]^R + \dots \tag{15}$$

Here objects $[A|A|\dots]$ should be computed by multiplying the matrices and concatenating the one-forms appearing in their entries into words within the bar construction. The operation $R$ reverses words,

$$[w_1|\dots|w_k]^R = [w_k|\dots|w_1]. \tag{16}$$

Because the matrix $A$ is nilpotent, the series in eq. 15 is guaranteed to terminate.

Using the matrix $T_A$, the symbol of a pair of forms $[\xi_i, \xi_j]$ is an entry of the matrix as follows:

$$\mathcal{S}([\xi_i, \xi_j]) = \langle \xi_i | T_A | \xi_j \rangle = (T_A)_{ji}. \tag{17}$$

Thus, each pair of forms in our basis is assigned an object in the bar construction. These objects are tensor product of one-forms, which may be more familiar: they are in essence our familiar notion of a symbol.

While one might worry that symbols of this sort are basis-dependent, in fact changes of basis consistent with the overall form in eq. 13 will leave the symbol invariant, provided we take into account relations on symbols implied by integration by parts. For the case of closed forms $\omega_i$, these relations read,

$$[\omega_1|\dots|\omega_k|df|\omega_{k+1}|\dots|\omega_n] = [\omega_1|\dots|\omega_k f|\omega_{k+1}|\dots|\omega_n] - [\omega_1|\dots|\omega_k|f\,\omega_{k+1}|\dots|\omega_n], \tag{18}$$

$$[df|\omega_1|\dots|\omega_n] = [f\,\omega_1|\dots|\omega_n] - f[\omega_1|\dots|\omega_n], \tag{19}$$

$$[\omega_1|\dots|\omega_n|df] = f[\omega_1|\dots|\omega_n] - [\omega_1|\dots|\omega_n f]. \tag{20}$$

Combining our two ingredients, this symbol map and the motivic coaction, naturally leads to a coaction of the following form:

$$\Delta \equiv (\mathrm{id} \otimes \mathcal{S})\Delta^{\mathrm{m}}. \tag{21}$$

Acting on the basis in eq. 12 with this coaction gives,

$$\Delta(I_a) = \sum_c I_c \otimes \mathcal{S}([\xi_c, \xi_a]). \tag{22}$$

As the symbol is only defined on unipotent quantities, this coaction as stated is as well. There is however a simple way it can be extended to consider more general quantities, at the cost of introducing an arbitrary choice.

In particular, a general motivic period $x$ can be decomposed into a bilinear combination of unipotent periods $u_i$ and semi-simple periods $s_i$ as follows:

$$x = \sum_i u_i s_i. \tag{23}$$

Semi-simple periods are defined to be periods for which the coaction acts trivially,

$$\Delta s = s \otimes 1, \tag{24}$$

while coactions act on unipotent periods according to eq. 22. This construction should be familiar from the polylogarithmic case, as it is the way the coaction handles powers of $\pi$: these cannot appear in the final entries as these are de Rham periods, but they can be consistently included in the first entry of the coaction.

There are in principle many legitimate choices for this decomposition, with different choices in general corresponding to inequivalent constructions of the symbol and coaction.[1] Readers should keep this in mind in what follows: while we make what we believe to be a reasonable choice in our constructions, there were many other possible choices.

## 2.3 Gauss-Manin system

In general, a basis of master integrals for a set of Feynman diagrams will satisfy a coupled system of differential equations [75–78]. This can be presented as a system of linear differential equations. Following ref. [74], we refer to this presentation as a Gauss-Manin system.

$$d\underline{I} = A\underline{I}. \tag{25}$$

---

[1]For example, it is always possible to include any unipotent period in the semi-simple part instead, in the extreme case trivializing the coaction completely.

If one instead focuses on the integrals in a particular sector, the system becomes inhomogeneous, with the inhomogeneity coming from contributions in lower sectors. Let us write the basis of master integrals in a single sector as,

$$\underline{J} = \{J_1, \ldots J_n\}. \tag{26}$$

We will have two cases of interest, the top sectors in the equal-mass and general-mass cases for the sunrise. For the equal-mass case at $l$ loops, $J_1 \ldots J_n$ correspond to the $l$ master integrals $J_{l,1}, J_{l,2}, \ldots J_{l,l}$ defined in eqs. 8 and 9. For the general-mass case, they instead correspond to the top sector integrals specified in eqs. 5 and 6. Denoting the relevant dimensionless kinematic ratios by $\underline{z}$ in either case, we have,

$$d\underline{J}(\underline{z}, \epsilon) = \mathbf{B}(\underline{z}, \epsilon)\underline{J}(\underline{z}, \epsilon) + \underline{N}(\underline{z}, \epsilon), \tag{27}$$

where $\underline{N}(\underline{z}, \epsilon)$ gives the inhomogeneous terms, which come from lower sectors. These terms vanish for the maximal cut, as such, the maximal cuts of top sector integrals satisfy the homogeneous form of this differential equation.[2] We are working with an $\epsilon$-regular basis of integrals, so all expressions here are finite in the $\epsilon \to 0$ limit. In particular, we write $\mathbf{B}(\underline{z}) = \mathbf{B}(\underline{z}, 0)$.

One can write a general solution to the homogeneous equation in terms of a matrix called the Wronskian,

$$d\mathbf{W}(\underline{z}) = \mathbf{B}(\underline{z})\mathbf{W}(\underline{z}). \tag{28}$$

Using the Wronskian, we can transform to a different basis, which will be a useful starting point in the following:

$$\underline{L}(\underline{z}, \epsilon) = \mathbf{W}(\underline{z})^{-1}\underline{J}(\underline{z}, \epsilon). \tag{29}$$

In this basis, the differential equation becomes,

$$d\underline{L}(\underline{z}, \epsilon) = \mathbf{W}(\underline{z})^{-1}\left[\mathbf{B}(\underline{z}, \epsilon) - \mathbf{B}(\underline{z})\right]\mathbf{W}(\underline{z}) + \mathbf{W}(\underline{z})^{-1}\underline{N}(\underline{z}, \epsilon), \tag{30}$$

which becomes especially simple when restricted to integer dimension with $\epsilon \to 0$, which we will do in the following:

$$d\underline{L}(\underline{z}) = \mathbf{W}(\underline{z})^{-1}\underline{N}(\underline{z}). \tag{31}$$

## 2.4 Frobenius basis and Griffiths transversality

In order to make the above differential equation explicit, one needs explicit representations for the inverse of the Wronskian $\mathbf{W}(\underline{z})^{-1}$ and for the inhomogeneities $\underline{N}(\underline{z})$. For the sunrise integrals the latter are simple, coming from tadpole integrals. The inverse of the Wronskian is potentially more complicated, and at higher loops one might expect it to contain higher degree polynomials in the maximal cuts. However, ref. [74] provides a few key simplifications in the specific case of the sunrise family, owing to their Calabi-Yau structure.

An alternative to the Gauss-Manin presentation represents a coupled system of differential equations in terms of a set of inhomogeneous higher-order differential equations satisfied by each master integral, called Picard-Fuchs differential equations

$$\mathcal{L}_k I_k(\underline{z}) = R_k(\underline{z}). \tag{32}$$

The inhomogeneous term is related to master integrals from lower sectors. Solutions of the homogeneous version (found by setting $R_k(\underline{z}) = 0$) close to a given point $\underline{z}_0$ can be characterized by indicial equations, polynomials whose roots give the leading exponents of $(z_i - z_{0,i})$

---

[2]This was first observed in the case of the two-loop sunrise integral in ref. [44]. It was shown in general in ref. [79], and elaborated in refs. [67, 80–82].

in a power series around $\underline{z}_0$. If these roots are degenerate then there exist logarithmic solutions, with higher powers of logarithms for higher degeneracy, organized in what is called a Frobenius basis.

If the differential equations describe the moduli space of a Calabi-Yau manifold then there is expected to be at least one point of maximal unipotent monodromy, or MUM point [72]. At a MUM point, all of the indicial roots are degenerate. There is then only one power series solution, accompanied by a hierarchical set of logarithmic solutions of increasing power. As such, it becomes quite straightforward to write down an ansatz for these solutions and solve for the Frobenius basis to the desired order around a MUM point. This Frobenius basis will serve as a particularly nice basis for the maximal cuts, and thus the entries of $\mathbf{W}(\underline{z})$

An additional property of Calabi-Yau manifolds of dimension $n$ is called Griffiths transversality. This property can be used to derive quadratic relations between elements of the Frobenius basis (and thus between maximal cuts and entries in $\mathbf{W}(\underline{z})$) of the following form:

$$\underline{\Pi}(\underline{z})^T \Sigma_l \partial_{\underline{z}}^{\underline{k}} \underline{\Pi}(\underline{z}) = \begin{cases} 0, & \text{for } 0 \le |\underline{k}| < n, \\ C_{\underline{k}}(\underline{z}), & \text{for } |\underline{k}| = n. \end{cases} \tag{33}$$

Here $\underline{\Pi}$ denotes a vector of Frobenius basis elements. $\Sigma_l$ is a constant matrix referred to as the intersection matrix. It can in general be derived straightforwardly, for example by checking this expression using the first few terms of series expansions of Frobenius basis elements for $0 \le |\underline{k}| < n$. The functions $C_{\underline{k}}(\underline{z})$ are often referred to as Yukawa couplings due to their role in Calabi-Yau compactifications. Ref. [74] describes how to find explicit expressions for these functions from the Picard-Fuchs differential equations and derives them for the equal-mass sunrise. We will review that derivation in section 3 and perform the analogous derivation for the generic-mass case in section 4. Once we have explicit expressions for the $C_{\underline{k}}(\underline{z})$, we can use the relations derived from Griffiths transversality to simplify the inverse of the Wronskian, representing its entries linearly in the Frobenius basis.

# 3 The equal-mass case

The goal of this section is to apply the equations of the previous section to the concrete example of the equal-mass master integral basis defined in eq. 6. We will begin with reviewing the maximal cuts of the equal-mass master integrals, as these are used in the construction of the Wronskian. We will also review the relations that can be found from Griffiths Transversality and in particular use these relations to simplify the inverse Wronskian.
Finally we will provide a recipe for defining a unipotent differential equation satisfied by these master integrals, which will in turn allow us to construct a symbol and coaction.

## 3.1 Frobenius basis

When all masses are equal the solution space of the Picard-Fuchs differential equation is spanned by $l-1$ logarithmic solutions containing up to $l$ powers of $\log(z)$. In the equal-mass case it is sufficient to consider only a single Picard-Fuchs operator of the type

$$\mathcal{L}_l = B_{l,l}(z)\partial_z^l + B_{l,l-1}(z)\partial_z^{l-1} + ... + B_{l,0}(z), \tag{34}$$

to generate the entire solution space. Such operators have been studied in [73] and a list of the relevant operators at different loop orders is provided. We look for solutions around the MUM-point $z_0 = 0$. At this MUM-point the solution space is spanned by the Frobenius basis

given by

$$\varpi_{l,k}(z) = \sum_{j=0}^{k} \frac{1}{(k-j)!} \log^{k-j}(z) \Sigma_{l,j}(z), \tag{35}$$

where $\Sigma_{l,k}(z)$ is a power series, normalized such that $\Sigma_{l,k}(z) = \delta_{k,0} z + \mathcal{O}(z^2)$.[3] For $k = 0$, we have

$$\varpi_{l,0}(z) = \Sigma_{l,0}(z) = \sum_{k_1,\dots,k_{l+1} \geq 0} \left( \frac{|k|!}{k_1! k_2! \dots k_{l+1}!} \right)^2 z^{|k|+1}, \tag{36}$$

where $|k| = \sum_{i}^{l+1} k_i$. The remaining $\Sigma_{l,k}(z)$'s can be derived by considering the fact that elements of the Frobenius basis satisfy the homogenous version of the Picard-Fuchs equation, i.e all elements satisfy $\mathcal{L}_l \varpi_{l,k}(z) = 0$. We collect all the Frobenius elements into a vector, which we denote as $\Pi_l(z) = (\varpi_{l,0}(z), \varpi_{l,1}(z), \dots, \varpi_{l,l-1}(z))$.

As mentioned earlier, Griffiths Transversality allows one to derive relations between elements of the Frobenius basis. We will in the following derive these relations, which we collectively refer to as *Griffiths Relations*. By using the explicit expressions of the equal-mass Frobenius basis in the context of Griffiths Transversality, it is possible to derive that the intersection matrix $\Sigma_l$ has the form

$$\Sigma_l = \begin{pmatrix} 0 & \dots & 0 & 1 \\ 0 & \dots & -1 & 0 \\ 0 & \cdot^{\cdot^{\cdot}} & 0 & 0 \\ (-1)^{l+1} & 0 & 0 & 0 \end{pmatrix}. \tag{37}$$

We now consider Griffiths Transversality in the case where $|k| = n = l-1$. Our goal is to derive a differential equation that is satisfied by the Yukawa couplings $C_l(z)$.

We start out by using the product rule to derive relations between first orders of derivatives with respect to the Frobenius basis:

$$C_l(z) = \partial_z(\underline{\Pi}_l(z)^T \Sigma_l \partial_z^{l-2} \underline{\Pi}(z)) - \partial_z \underline{\Pi}_l(z)^T \Sigma_l \partial_z^{l-2} \underline{\Pi}_l(z). \tag{38}$$

The first term is zero, due to eq. 33. We can continue using the product rule this way and find a more generalized version of eq. 33:

$$C_l(z) = (-1)^k \partial_z^k \underline{\Pi}_l(z)^T \Sigma_l \partial_z^{l-1-k} \underline{\Pi}_l(z). \tag{39}$$

If one then differentiates the expression in eq. 33 and uses the product rule successively, one finds,

$$\begin{aligned}
\underline{\Pi}_l(z)^T \Sigma_l \partial_z^l \underline{\Pi}_l(z) &= \partial_z C_l(z) - \partial_z \underline{\Pi}_l(z)^T \Sigma_l \partial_z^{l-1} \underline{\Pi}_l(z) \\
&= l \partial_z C_l(z) + (-1)^l \partial_z^l \underline{\Pi}_l(z)^T \Sigma_l \underline{\Pi}_l(z).
\end{aligned} \tag{40}$$

Due to the $(-1)^{l+1}$ symmetry of $\Sigma_l$, we can reduce this to

$$\underline{\Pi}_l(z)^T \Sigma_l \partial_z^l \underline{\Pi}_l(z) = \frac{l}{2} \partial_z C_l(z). \tag{41}$$

So far we have only developed relations between different degrees of differential orders of the Frobenius basis. To find the differential equation whose solution is the Yukawa-coupling we make use of the fact that the Picard-Fuchs operator is an $l$-order differential equation that annihilates the Frobenius basis:

$$\begin{aligned}
0 &= \underline{\Pi}_l(z)^T \Sigma_l \mathcal{L}_l \underline{\Pi}_l(z) \\
&= B_{l,l-1}(z) \underline{\Pi}_l(z)^T \Sigma_l \partial_z^{l-1} \underline{\Pi}_l(z) + B_{l,l}(z) \underline{\Pi}_l(z)^T \Sigma_l \partial_z^l \underline{\Pi}_l(z).
\end{aligned} \tag{42}$$

---

[3]And not to be confused with the intersection matrix $\Sigma_l$.

We can now use the relations we have found above, namely eq. 33 and eq. 41, to find a differential equation that is satisfied by $C_l(z)$

$$\partial_z C_l(z) + \frac{2}{l}\frac{B_{l,l-1}(z)}{B_{l,l}(z)}C_l(z) = 0. \tag{43}$$

This differential equation is solved by,

$$C_l(z) = \frac{1}{z^{l-3}\prod_{k\in\Delta^{(l)}}(1-kz)}, \tag{44}$$

where

$$\Delta^{(l)} = \bigcup_{j=0}^{\lceil\frac{l-1}{2}\rceil}\left\{(l+1-2j)^2\right\}. \tag{45}$$

We can also use eq. 40 to generate a series of relations through differentiation. To begin, we see that we can rewrite eq. 40, by using the Picard Fuchs operator

$$\begin{aligned}
\partial_z C_l(z) &= \underline{\Pi}_l(z)^T\Sigma_l\partial_z^l\underline{\Pi}_l(z) + \partial_z\underline{\Pi}_l(z)^T\Sigma_l\partial_z^{l-1}\underline{\Pi}_l(z)\\
&= -\sum_{j=1}^{l-1}\frac{B_{l,j}(z)}{B_{l,l}(z)}\underline{\Pi}_l(z)^T\Sigma_l\partial_z^j\underline{\Pi}_l(z) + \partial_z\underline{\Pi}_l(z)^T\Sigma_l\partial_z^{l-1}\underline{\Pi}_l(z).
\end{aligned} \tag{46}$$

Due to Griffiths relations, we realize that the only non-vanishing term in the sum is $j = l-1$. We can then use the differential equation 43 to collect terms. This gives us even more relations between elements of the Frobenius basis

$$\left(1-\frac{l}{2}\right)\partial_z C_l(z) = \partial_z\underline{\Pi}_l(z)^T\Sigma_l\partial_z^{l-1}\underline{\Pi}_l(z). \tag{47}$$

We can proceed in this fashion, i.e taking the derivative of eq. 41 and using the Picard-Fuchs operator to re-write the $l$-order differential. Eventually we will have enough relations to calculate all entries in the matrix

$$\mathbf{Z}_l(z) = \begin{pmatrix} \underline{\Pi}(z)^T\Sigma_l\underline{\Pi}(z) & \cdots & \underline{\Pi}(z)^T\Sigma_l\partial_z^{l-1}\underline{\Pi}(z)\\ \vdots & \ddots & \vdots\\ \partial_z^{l-1}\underline{\Pi}(z)^T\Sigma_l\underline{\Pi}(z) & \cdots & \partial_z^{l-1}\underline{\Pi}(z)^T\Sigma_l\partial_z^{l-1}\underline{\Pi}(z) \end{pmatrix}. \tag{48}$$

In terms of $\mathbf{Z}_l(z)$, it is possible to express the inverse Wronskian as

$$\mathbf{W}_l(z)^{-1} = \Sigma_l\mathbf{W}_l(z)^T\mathbf{Z}_l(z)^{-1}. \tag{49}$$

It has a $(-1)^{l+1}\mathbf{Z}_l(z) = \mathbf{Z}_l(z)^T$ symmetry. We explicitly show the result for the $l$'th column of the inverse Wronskian, with $1 \le k \le l$, as this will be of great use for us later on,

$$W(z)_{k,l}^{-1} = \frac{(-1)^{l+k}\varpi_{l,l-k}(z)}{C_l(z)}. \tag{50}$$

We once again emphasise the importance of the Griffiths relations, as highlighted in ref. [74]. Our results here are be linear in elements of the Frobenius basis. Had we instead inverted the Wronskian as if it was a generic matrix, we would be left with degree $(l-1)$ polynomials of elements in the Wronskian.

### 3.2 Unipotent differential equation

As mentioned in the introduction to this section, the construction of a symbol and coaction on our master integrals begins with eq. 31. We need only find the inhomogeneous term $\underline{N}_l(z)$ to write the differential equation explicitly. This term can be found by realising the connection between the Picard-Fuchs equation and the Gauss-Manin equation. In particular, acting with the Picard-Fuchs operator on the top sector yields

$$\mathcal{L}_l \underline{J}_l(z) = \underline{N}_l(z). \tag{51}$$

In the equal-mass case, this inhomogeneity has the form

$$\underline{N}_l(z) = \left(0, ..., 0, (-1)^{l+1}(l+1)! \frac{z}{z^l \prod_{k \in \Delta^{(l)}}(1-kz)}\right), \tag{52}$$

where $\Delta^{(l)}$ is the same as eq. 45.

We can now combine eq. 50 and eq. 52 to explicitly write the differential equation as

$$dL_{l,k}(z) = (-1)^{k+1} \frac{(l+1)!}{z^2} \varpi_{l,k-1}(z)dz, \quad \text{for } 1 \le k \le l. \tag{53}$$

We will now need to decompose this equation into a unipotent and semi-simple part. As explained in section 2, there are many choices of doing so. We choose to define the unipotent period[4]

$$\tau_j(z) = \frac{\varpi_{l,j}(z)}{\varpi_{l,0}(z)}, \quad \le j \le l-1. \tag{54}$$

Writing the differential equation in terms of unipotent and semi-simple periods we obtain

$$dL_{l,k}(z) = (-1)^{k+1} \frac{(l+1)!}{z^2} \varpi_{l,0}(z)\tau_{k-1}(z)dz. \tag{55}$$

It has previously been mentioned that in order to build a coaction and symbol, we need our master integrals to satisfy a unipotent differential equation

$$d\underline{I}_l(z) = \mathbf{N}(z)\underline{I}_l(z), \tag{56}$$

where $\mathbf{N}(z)$ is a nilpotent matrix. The master integrals in eq. 55 do not satisfy a unipotent differential equation of this type. We can instead define a larger basis, which satisfies a unipotent differential equation as well as containing the master integrals. We define the expanded $L$-basis as

$$\underline{T}_l(z) = \left(L_{l,1}(z), ..., L_{l,l}(z), \tau_1(z), ..., \tau_{l-1}(z), 1\right). \tag{57}$$

This basis will satisfy a unipotent differential equation and enable us to work out a coaction and symbol on the master integrals. For example, at $l = 2$ we have the basis

$$\underline{T}_l(z) = \left(L_{2,1}(z), L_{2,2}(z), \tau_1(z), 1\right), \tag{58}$$

which satisfies the unipotent differential equation

$$d\begin{pmatrix} L_{2,1}(z) \\ L_{2,2}(z) \\ \tau_1(z) \\ 1 \end{pmatrix} = \begin{pmatrix} 0 & 0 & 0 & \frac{3!}{z^2}\varpi_{2,0}(z)dz \\ 0 & 0 & -\frac{3!}{z^2}\varpi_{2,0}(z)dz & 0 \\ 0 & 0 & 0 & d\tau_1 \\ 0 & 0 & 0 & 0 \end{pmatrix} \begin{pmatrix} L_{2,1}(z) \\ L_{2,2}(z) \\ \tau_1(z) \\ 1 \end{pmatrix}, \tag{59}$$

---

[4]Note that this is the inverse of the definition present in version one of this paper on the arXiv. Both definitions give valid coactions and symbols, but this definition leads to more useful series expansions and we believe this outweighs any confusion caused by the shift between versions.

where we have used eq. 55 to find the entries of the matrix. We can perform this procedure for arbitrary loop order, finding a $2l \times 2l$ matrix with entries

$$
\mathbf{N}_l(z)_{i,j} = \begin{cases} (-1)^{i+1} \frac{(l+1)!}{z^2} \varpi_{l,0}(z) dz, & \text{for } i+j = 2l+1, \quad 1 \le i \le l, \quad l < j \le 2l, \\ d\tau_{i-l}, & \text{for } j = 2l, \quad l < i < 2l, \\ 0, & \text{otherwise.} \end{cases} \tag{60}
$$

It can be shown that for any loop order $l$, $\mathbf{N}_l(z)$ has the property $\mathbf{N}_l(z)^3 = 0$, and thus is indeed nilpotent.

We are particularly interested in the coaction and symbol for loop orders $l \ge 4$, as this is the order at which the master integrals can no longer be expressed in terms of elliptic functions. As an example, we present the nilpotent matrix for $l = 4$:

$$
\mathbf{N}_4(z) = \begin{pmatrix} 0 & 0 & 0 & 0 & 0 & 0 & 0 & \frac{5!}{z^2} \varpi_{4,0}(z) dz \\ 0 & 0 & 0 & 0 & 0 & 0 & -\frac{5!}{z^2} \varpi_{4,0}(z) dz & \\ 0 & 0 & 0 & 0 & 0 & \frac{5!}{z^2} \varpi_{4,0}(z) dz & 0 & 0 \\ 0 & 0 & 0 & 0 & -\frac{5!}{z^2} \varpi_{4,0}(z) dz & 0 & 0 & 0 \\ 0 & 0 & 0 & 0 & 0 & 0 & 0 & d\tau_1 \\ 0 & 0 & 0 & 0 & 0 & 0 & 0 & d\tau_2 \\ 0 & 0 & 0 & 0 & 0 & 0 & 0 & d\tau_3 \\ 0 & 0 & 0 & 0 & 0 & 0 & 0 & 0 \end{pmatrix}. \tag{61}
$$

The structure is noticeably similar to that at two loops.

### 3.3 Symbol

We will now use our nilpotent matrix in eq. 60 to build a symbol operator. The first step is to define the unipotent matrix $T_N$

$$
T_N = 1 + [\mathbf{N}_l]^R + [\mathbf{N}_l|\mathbf{N}_l]^R + [\mathbf{N}_l|\mathbf{N}_l|\mathbf{N}_l]^R + \dots, \tag{62}
$$

where we have suppressed $z$-dependence. Using eq. 60, we can derive a general construction of $[\mathbf{N}_l|\mathbf{N}_l]^R$ and $[\mathbf{N}_l|\mathbf{N}_l|\mathbf{N}_l]^R$

$$
[\mathbf{N}_l|\mathbf{N}_l]_{ij}^R = \begin{cases} [d\tau_j | (-1)^{j+1} \frac{(l+1)!}{z^2} \varpi_{l,0}(z) dz], & \text{for } 2 \le j \le l, \quad i = 2l, \\ 0, & \text{otherwise,} \end{cases} \tag{63}
$$

$$
[\mathbf{N}_l|\mathbf{N}_l|\mathbf{N}_l]_{ij}^R = 0.
$$

We now define the symbol on a pair of periods $[\xi_i, \xi_j]$, where $\xi_i, \xi_j \in \underline{T}_l(z)$, as

$$
\mathcal{S}([\xi_i, \xi_j]) = (T_N)_{ji}. \tag{64}
$$

To see how this works in practice, consider the example of $l = 2$. In this case, the non-zero symbols are

$$
\mathcal{S}([\xi_i, \xi_i]) = 1,
$$

$$
\mathcal{S}([\tau_1, L_{2,2}(z)]) = \left[ -\frac{3!}{z^2} \varpi_{2,0}(z) dz \right],
$$

$$
\mathcal{S}([1, L_{2,1}(z)]) = \left[ -\frac{3!}{z^2} \varpi_{2,0}(z) dz \right], \tag{65}
$$

$$
\mathcal{S}([1, L_{2,2}(z)]) = \left[ d\tau_1 | \frac{3!}{z^2} \varpi_{2,0}(z) dz \right],
$$

$$
\mathcal{S}([1, \tau_1(z)]) = [d\tau_1].
$$

Symbols of the form $\mathcal{S}([1,\xi_j])$ are the type we would usually refer to as $\mathcal{S}(\xi_j)$ in the poly-logarithmic case, the "symbol of the function". The other type of pair $\mathcal{S}([\tau_1,L_{2,2}(z)])$ has a different interpretation. It may be viewed as an integral of the form defining $L_{2,2}(z)$ over a contour that encircles all of $\tau_1$'s singularities.[5]

We delay a comparison of these expressions to the two-loop elliptic symbols in the literature to the next subsection.

We can similarly work out the symbol in the more interesting case of $l = 4$, where the master integrals are no longer expressible in terms of elliptic functions. In this case we find the non-zero symbols to be

$$
\begin{aligned}
\mathcal{S}([\xi_i,\xi_i]) &= 1\,, \\
\mathcal{S}([\tau_1(z),L_{4,4}(z)]) &= \left[\frac{5!}{z^2}\varpi_{4,0}(z)dz\right]\,, \\
\mathcal{S}([\tau_2(z),L_{4,3}(z)]) &= -\left[\frac{5!}{z^2}\varpi_{4,0}(z)dz\right]\,, \\
\mathcal{S}([\tau_3(z),L_{4,2}(z)]) &= \left[\frac{5!}{z^2}\varpi_{4,0}(z)dz\right]\,, \\
\mathcal{S}([1,L_{4,1}(z)]) &= -\left[\frac{5!}{z^2}\varpi_{4,0}(z)dz\right]\,, \\
\mathcal{S}([1,L_{4,2}(z)]) &= \left[d\tau_3|\frac{5!}{z^2}\varpi_{4,0}(z)dz\right]\,, \\
\mathcal{S}([1,L_{4,3}(z)]) &= -\left[d\tau_2|\frac{5!}{z^2}\varpi_{4,0}(z)dz\right]\,, \\
\mathcal{S}([1,L_{4,4}(z)]) &= \left[d\tau_1|\frac{5!}{z^2}\varpi_{4,0}(z)dz\right]\,, \\
\mathcal{S}([1,\tau_1(z)]) &= [d\tau_1]\,, \\
\mathcal{S}([1,\tau_2(z)]) &= [d\tau_2]\,, \\
\mathcal{S}([1,\tau_3(z)]) &= [d\tau_3]\,.
\end{aligned}
\tag{66}
$$

By first inspection these all appear to be of length two. This may be surprising, as naively one might expect higher-length functions to appear at higher loop orders, as polylogarithmic functions at higher orders are often of higher transcendental weight. We will comment on this surprise in the next subsection, and offer a partial explanation.

## 3.4 Coaction

Now that we have defined symbols on all pairs of elements in $\underline{T}_l(z)$, we can use

$$
\Delta(T_{l,k}) = \sum_{i=1}^{2l}\left(T_{l,i}(z)\otimes\mathcal{S}([\xi_i,\xi_k])\right)\,,
\tag{67}
$$

to calculate the coaction on elements in this basis.

We again start with $l = 2$ example. In this case we find

$$
\begin{aligned}
\Delta(L_{2,1}(z)) &= L_{2,1}(z)\otimes 1 - 1\otimes\left[\frac{3!}{z^2}\varpi_{2,0}(z)dz\right]\,, \\
\Delta(L_{2,2}(z)) &= L_{2,2}(z)\otimes 1 + 1\otimes\left[d\tau_1|\frac{3!}{z^2}\varpi_{2,0}(z)dz\right] + \tau_1(z)\otimes\left[\frac{3!}{z^2}\varpi_{2,0}(z)dz\right]\,, \\
\Delta(\tau_1(z)) &= \tau_1\otimes 1 + 1\otimes[d\tau_1]\,.
\end{aligned}
\tag{68}
$$

---

[5]Note that this is only an analogy, as de Rham pairs are not actually integrals.

As we know how the coaction acts on every element of $\underline{T}_l(z)$, it is possible to calculate the coaction of our original master integrals in $\underline{J}_l(z)$. Remember that the two are related by $\underline{J}_l(z) = \mathbf{W}_l(z)\underline{L}_l(z)$, meaning that we can express elements of $\underline{J}_l(z)$ in terms of elements in $\underline{L}_l(z)$ and $\mathbf{W}_l(z)$. For instance, in the two-loop example, we have that

$$
\begin{aligned}
J_{2,1}(z) &= L_{2,1}(z)\varpi_{2,0}(z)\tau_1(z) + L_{2,2}(z)\varpi_{2,0}(z), \\
J_{2,2}(z) &= L_{2,1}(z)\big[\varpi_{2,0}(z)\partial_z\tau_1(z) + \tau_1(z)\partial_z\varpi_{2,0}(z)\big] + L_{2,2}\partial_z\varpi_{2,0}(z).
\end{aligned}
\tag{69}
$$

To find the coaction on these quantities, we make use of the compatibility requirement between multiplication and coaction:

$$
\Delta(T_{l,k}(z)T_{l,j}(z)) = \Delta(T_{l,k}(z))\cdot\Delta(T_{l,j}(z)),
\tag{70}
$$

as well as the coaction's action on semi-simple quantities such as $\varpi_{2,0}(z)$ here, described generically in eq. 24. Doing this, we obtain,

$$
\begin{aligned}
\Delta(J_{2,1}(z)) &= J_{2,1}(z)\otimes 1 + \varpi_{2,0}(z)\left(L_{2,1}(z)\otimes[d\tau_1] + 1\otimes\left[d\tau_1|\frac{3!}{z^2}\varpi_{2,0}(z)dz\right]\right), \\
\Delta(J_{2,2}(z)) &= J_{2,2}(z)\otimes 1 + \partial_z\varpi_{2,0}(z)\left(L_{2,1}(z)\otimes[d\tau_1] + 1\otimes\left[d\tau_1|\frac{3!}{z^2}\varpi_{2,0}(z)dz\right]\right),
\end{aligned}
\tag{71}
$$

where we have used the relations in eq. 19 and 20 as well as tensor identities for multiplication to simplify the results. Here the broad structure (in particular, the position of factors $d\tau$) resembles the coactions and symbols found by Broedel et. al using their construction of a coaction and symbol for elliptic multiple polylogarithms [62]. However, despite this broad resemblance, it is important to note that the objects appearing in our coaction are distinct from theirs, as the prefactors we remove differ by a factor of $m^2/(m^2 - p^2)$.

We now move on to calculate the coaction at higher loop orders. We consider the case $l = 4$ and obtain the following results:

$$
\begin{aligned}
\Delta(\tau_k(z)) &= \tau_k\otimes 1 + 1\otimes[d\tau_k], \\
\Delta(L_{4,1}(z)) &= L_{4,1}(z)\otimes 1 - 1\otimes\left[\frac{5!}{z^2}\varpi_{4,0}(z)dz\right], \\
\Delta(L_{4,2}(z)) &= L_{4,2}(z)\otimes 1 + 1\otimes\left[d\tau_3|\frac{5!}{z^2}\varpi_{4,0}(z)dz\right] + \tau_3(z)\otimes\left[\frac{5!}{z^2}\varpi_{4,0}(z)dz\right], \\
\Delta(L_{4,3}(z)) &= L_{4,3}(z)\otimes 1 - 1\otimes\left[d\tau_2|\frac{5!}{z^2}\varpi_{4,0}(z)dz\right] - \tau_2(z)\otimes\left[\frac{5!}{z^2}\varpi_{4,0}(z)dz\right], \\
\Delta(L_{4,4}(z)) &= L_{4,4}(z)\otimes 1 + 1\otimes\left[d\tau_1|\frac{5!}{z^2}\varpi_{4,0}(z)dz\right] + \tau_1(z)\otimes\left[\frac{5!}{z^2}\varpi_{4,0}(z)dz\right].
\end{aligned}
\tag{72}
$$

We include MATHEMATICA code to generate these coactions through five loops in an ancillary file, `SunriseSupplementaryMaterial.nb`, in the arXiv version of this publication, along with code to generate the symbols from the previous section.

The structure is similar to that of the $l = 2$ case. Again it is possible to calculate the coaction for the $\underline{J}_4(z)$ basis using the same procedure as for the $l = 2$ example. For instance the coaction of the first element in $\underline{J}_4(z)$ is

$$
\begin{aligned}
\Delta(J_{4,1}(z)) = J_{4,1}(z)\otimes 1 + \varpi_{4,0}(z)\bigg(&L_{4,1}(z)\otimes[d\tau_1] \\
&+ \tau_2\otimes\left[d\tau_3|\frac{5!}{z^2}\varpi_{4,0}(z)dz\right] + L_{4,2}(z)\otimes[d\tau_2] + 1\otimes\left[d\tau_3|d\tau_2|\frac{5!}{z^2}\varpi_{4,0}(z)dz\right] \\
&- \tau_3\otimes\left[d\tau_2|\frac{5!}{z^2}\varpi_{4,0}(z)dz\right] + L_{4,3}(z)\otimes[d\tau_3] - 1\otimes\left[d\tau_2|d\tau_3|\frac{5!}{z^2}\varpi_{4,0}(z)dz\right] \\
&+ 1\otimes\left[d\tau_1|\frac{5!}{z^2}\varpi_{4,0}(z)dz\right]\bigg).
\end{aligned}
\tag{73}
$$

This expression should partially reassure us. If we think of the symbol of the function $J_{4,1}(z)$ as encoded in terms of the coaction of the form $1 \otimes X$, then the symbols present in these terms have length three, not two. This property is also present at three loops, where it agrees with the length of the elliptic multiple polylogarithm functions used to represent the equal-mass banana graph at this order in ref. [68]. While this partial agreement with the literature is reassuring, it may be strange that the length of our symbols continues to be three at four loops. In fact, the symbols obtained from our formalism for the master integrals $J_{l \geq 3, i}$ will *always* be of length three, at any loop order, contrary to the expectation that the length of functions increases at higher orders in perturbation theory along with their transcendental weight.

One perspective on the situation is as follows: It was explained earlier in section 2.2 that there is considerable freedom in choosing a basis of functions when defining a symbol. Indeed it was not needed to expand the $\underline{L}_l(z)$-basis with $l$ elements of $\tau_k(z)$ to define a basis that satisfies a unipotent differential equation. We could have simply expanded the $\underline{L}_l(z)$ basis with a simple "1". Had we done so, all symbols would be of length one and no new information would be obtained. Due to the arbitrariness of decomposing functions into semi-simple and unipotent parts, and in adding new elements to the basis, it is quite likely that there exists another choice, in which the symbol length is even longer than that which we found.[6] It is therefore important to note that even though we have found symbol lengths of three for $l > 2$, this should be thought of as an artifact of choosing our particular basis and its decomposition of semi-simple and unipotent periods.

## 4 The generic-mass case

We now move to the generic-mass case. We will in this section attempt to generalize the results of the previous section. We will therefore consider the $\epsilon = 0$ reduced basis of master integrals expressed in eq. 6. Many of the expressions we had in the equal-mass case are in the generic-mass case replaced by very long and complicated polynomials in all $l + 1$ scales. As we are mostly interested in the structure of the symbol and coaction, we will in general not derive these expressions explicitly, but instead describe how they may be derived.

### 4.1 Frobenius basis

It is possible to extend the one-dimensional Frobenius basis in eq. 35 to the multivariate case. In this case we consider a set of differential operators $\mathcal{D} = \{\mathcal{L}_1, ..., \mathcal{L}_s\}$. The solution space is spanned by the set of linearly independent functions $f(\underline{z})$ that are simultaneously annihilated by all of the operators in $\mathcal{D}$. This set of differential operators is by no means unique, in the sense that two different sets of operators may generate the exact same solution space [74]. We are looking for operators of the form

$$\mathcal{L}_{l,k} = \sum_{\underline{p}}^{|\underline{p}|=l} \beta_{k,\underline{p}}(\underline{z}) \partial_{\underline{z}}^{\underline{p}}, \tag{74}$$

where $p$ is an $l + 1$ vector to generate our solution space. This particular form of operators is important as it allows us to relate an $l$-order differential to lower order differentials, which is needed for the derivation of generic-mass Griffiths relations.

---

[6]An example of such a choice appeared after our initial arXiv publication in refs. [83,84] Ref. [85] additionally expresses ratios of maximal cuts corresponding to our $\tau_k(z)$ as iterated integrals, making the connection between the formalisms directly manifest.

It is again possible to find a solution around a MUM-point, located at $\underline{z}_0 = 0$. Around this point there exists one power series type solution

$$\varpi_0(\underline{z}) = \sum_{j_1,\ldots,j_{l+1}\geq 0} \left(\frac{|j|!}{j_1!\ldots j_{l+1}!}\right)^2 \prod_{n=1}^{l+1} z_n^{j_n}. \tag{75}$$

The rest of the solutions are graded by increasing orders of logarithms. In the generic-mass case however, there are typically multiple distinct solutions at each order. For instance, solutions for $l > 2$ have $l + 1$ single-logarithmic solutions, normalized as

$$\varpi_1^k(\underline{z}) = \log(z_k) + \mathcal{O}(\underline{z}). \tag{76}$$

In total one finds that there are $\lambda = 2^{l+1} - \binom{l+2}{\lfloor\frac{l+2}{2}\rfloor}$ elements in the Frobenius basis. For all loop orders there exists only one solution with total logarithmic order $l - 1$, which we denote as $\varpi_\lambda(\underline{z})$ [73].

The elements in the generic-mass Frobenius basis are also related through Griffiths Transversality (eq. 33). Just as in the equal-mass case, this can be used to simplify the inverse Wronskian. We are therefore motivated to find generic-mass version of the Griffiths relations derived in section 2.4.

The intersection matrix $\Sigma_l$ in the generic-mass case has the same $(-1)^{l+1}$ symmetry as in the equal-mass case. The construction of the intersection matrix is however not the same. Using the expansions for the generic-mass Frobenius basis one can derive that the three and four loop intersection matrices have the forms:

$$\Sigma_3 = \begin{pmatrix} 0 & 0 & 0 & 0 & 0 & 1 \\ 0 & 0 & -1 & -1 & -1 & 0 \\ 0 & -1 & 0 & -1 & -1 & 0 \\ 0 & -1 & -1 & 0 & -1 & 0 \\ 0 & -1 & -1 & -1 & 0 & 0 \\ 1 & 0 & 0 & 0 & 0 & 0 \end{pmatrix}, \tag{77}$$

$$\Sigma_4 = \begin{pmatrix} 0 & 0 & 0 & 0 & 0 & 0 & 0 & 0 & 0 & 0 & 0 & 1 \\ 0 & 0 & 0 & 0 & 0 & 0 & -1 & 0 & 0 & 0 & 0 & 0 \\ 0 & 0 & 0 & 0 & 0 & 0 & 0 & -1 & 0 & 0 & 0 & 0 \\ 0 & 0 & 0 & 0 & 0 & 0 & 0 & 0 & -1 & 0 & 0 & 0 \\ 0 & 0 & 0 & 0 & 0 & 0 & 0 & 0 & 0 & -1 & 0 & 0 \\ 0 & 0 & 0 & 0 & 0 & 0 & 0 & 0 & 0 & 0 & -1 & 0 \\ 0 & 1 & 0 & 0 & 0 & 0 & 0 & 0 & 0 & 0 & 0 & 0 \\ 0 & 0 & 1 & 0 & 0 & 0 & 0 & 0 & 0 & 0 & 0 & 0 \\ 0 & 0 & 0 & 1 & 0 & 0 & 0 & 0 & 0 & 0 & 0 & 0 \\ 0 & 0 & 0 & 0 & 1 & 0 & 0 & 0 & 0 & 0 & 0 & 0 \\ 0 & 0 & 0 & 0 & 0 & 1 & 0 & 0 & 0 & 0 & 0 & 0 \\ -1 & 0 & 0 & 0 & 0 & 0 & 0 & 0 & 0 & 0 & 0 & 0 \end{pmatrix}. \tag{78}$$

From now on we will assume that $n = |\underline{k}| = l - 1$ in eq. 33 and derive relations from that. Just as in the equal-mass case, we can gain more relations by using the product rule. However, since $\underline{k}$ is now a vector, we have a choice of which derivative to "pull out" of $\partial_{\underline{z}}^{\underline{k}}$. We can keep our notation general by pulling out $\partial_{z_i}$, as long as the restriction to cases where $\underline{k}_i \geq 1$ is understood. By pulling out one $\partial_{z_i}$ we get the equation

$$C_{\underline{k}}(\underline{z}) = \partial_{z_i}\left(\underline{\Pi}(\underline{z})\Sigma_l\partial_{\underline{z}}^{k-1_i}\underline{\Pi}(\underline{z})\right) - \partial_{z_i}\underline{\Pi}(\underline{z})\Sigma_l\partial_{\underline{z}}^{k-1_i}\underline{\Pi}(\underline{z}), \tag{79}$$

where $\underline{k} - 1_i$, means that we are subtracting one from the $i$'th entry in $\underline{k}$. Equivalently one can think of $1_i = (0, ..., 0, 1, 0, ..., 0)$ as an $l+1$ vector, where the $i$ indicates the entry that has the "1". The first term on the right hand side is zero due to the fact that $|k - 1_i| < l - 1$. We can once again pull out a derivative $\partial_{z_j}$ from the second term in eq. 79, where $j$ need not be different from $i$

$$C_{\underline{k}}(\underline{z}) = -\partial_{z_j}\partial_{z_i}\left(\partial_{z_i}\underline{\Pi}(\underline{z})\Sigma_l\partial_{\underline{z}}^{k-1_i-1_j}\underline{\Pi}(\underline{z})\right) + \partial_{z_j}\partial_{z_i}\underline{\Pi}(\underline{z})\Sigma_l\partial_{\underline{z}}^{k-1_i-1_j}\underline{\Pi}(\underline{z}). \tag{80}$$

It is once again possible to show that the first term on the right hand side vanishes due to eq. 33. We can keep going in this way, and in general we obtain

$$C_{\underline{k}}(\underline{z}) = \partial_{\underline{z}}^s\underline{\Pi}(\underline{z})\Sigma_l\partial_{\underline{z}}^{k-s}\underline{\Pi}(\underline{z}), \tag{81}$$

where $|s| \leq |k|$ and $k_i - s_i \geq 0$ for all $1 \leq i \leq l+1$.

We will now derive the next set of Griffiths relations, by taking the $i$'th derivative of eq. 33

$$\partial_{z_i}\left(\underline{\Pi}(\underline{z})\Sigma_l\partial_{\underline{z}}^k\underline{\Pi}(\underline{z})\right) = \partial_{z_i}C_{\underline{k}}(\underline{z}),$$
$$\underline{\Pi}(\underline{z})\Sigma_l\partial_{\underline{z}}^{k+1_i}\underline{\Pi}(\underline{z}) = \partial_{z_i}C_{\underline{k}}(\underline{z}) - \partial_{z_i}\underline{\Pi}(\underline{z})\Sigma_l\partial_{\underline{z}}^k\underline{\Pi}(\underline{z}). \tag{82}$$

If we now pull out an $\partial_{z_j}$ from $\partial_{\underline{z}}^k$ in the second term, we find

$$\underline{\Pi}(\underline{z})\Sigma_l\partial_{\underline{z}}^{k+1_i}\underline{\Pi}(\underline{z}) = \partial_{z_i}C_{\underline{k}}(\underline{z}) - \partial_{z_j}\left(\partial_{z_i}\underline{\Pi}(\underline{z})\Sigma_l\partial_{\underline{z}}^{k-1_j}\underline{\Pi}(\underline{z})\right) + \partial_{z_j}\partial_{z_i}\underline{\Pi}(\underline{z})\Sigma_l\partial_{\underline{z}}^{k-1_j}\underline{\Pi}(\underline{z})$$
$$= \partial_{z_i}C_{\underline{k}}(\underline{z}) - \partial_{z_j}C_{\underline{k}+1_i-1_j}(\underline{z}) + \partial_{z_j}\partial_{z_i}\underline{\Pi}(\underline{z})\Sigma_l\partial_{\underline{z}}^{k-1_j}\underline{\Pi}(\underline{z}) \tag{83}$$

(We must still be careful to apply this only in cases where $\underline{k}_j - 1_j \geq 0$). Performing this procedure successively, we find

$$\underline{\Pi}(\underline{z})\Sigma_l\partial_{\underline{z}}^{k+1_i}\underline{\Pi}(\underline{z}) = \partial_{z_i}C_{\underline{k}}(\underline{z}) + \sum_{m=1}^{l+1}(-1)^{|\underline{k}_m+1|}\underline{k}_m\partial_{z_m}C_{\underline{k}+1_i-1_m}(\underline{z}) + (-1)^{|\underline{k}+1_i|}\partial_{\underline{z}}^k\underline{\Pi}(\underline{z})\Sigma_l\underline{\Pi}(\underline{z}). \tag{84}$$

Due to the $(-1)^{l+1}$ symmetry of $\Sigma_l$ we can write 84 as

$$\underline{\Pi}(\underline{z})\Sigma_l\partial_{\underline{z}}^{k+1_i}\underline{\Pi}(\underline{z}) = \frac{1}{2}\partial_{z_i}C_{\underline{k}}(\underline{z}) - \sum_{m=1}^{l+1}(-1)^{k_m}\frac{k_m}{2}\partial_{z_m}C_{\underline{k}+1_i-1_m}(\underline{z}). \tag{85}$$

We can insert eq. 85 back into eq. 82 to find yet more relations

$$\partial_{z_i}\underline{\Pi}(\underline{z})\Sigma_l\partial_{\underline{z}}^k\underline{\Pi}(\underline{z}) = \frac{1}{2}\partial_{z_i}C_{\underline{k}}(\underline{z}) + \sum_{m=1}^{l+1}(-1)^{k_m}\frac{k_m}{2}\partial_{z_m}C_{\underline{k}+1_i-1_m}(\underline{z}). \tag{86}$$

We will now derive a generic-mass version of the differential equation in eq. 43. To do this we again use the fact that the Picard-Fuchs operators annihilate the Frobenius basis

$$0 = \underline{\Pi}(\underline{z})\Sigma_l\mathcal{L}_{l,\underline{k}}\underline{\Pi}(\underline{z})$$
$$= \underline{\Pi}(\underline{z})\Sigma_l\beta_{l,\underline{k}+1_i}^{(l)}\partial_{\underline{z}}^{k+1_i}\underline{\Pi}(\underline{z}) + \underline{\Pi}(\underline{z})\Sigma_l\beta_{l,\underline{k}}^{(l)}\partial_{\underline{z}}^k\underline{\Pi}(\underline{z}). \tag{87}$$

Combining this with eq. 33 and 85 we get

$$\sum_{m=1}^{l+1}(-1)^{k_m}\frac{k_m}{2}\partial_{z_m}C_{\underline{k}+1_i-1_m}(\underline{z}) = \frac{1}{2}\partial_{z_i}C_{\underline{k}}(\underline{z}) + \frac{\beta_{l,\underline{k}}^{(l)}(\underline{z})}{\beta_{l,\underline{k}+1_i}^{(l)}}C_{\underline{k}}(\underline{z}). \tag{88}$$

This differential equation may be solved iteratively, but would be somewhat of a distraction from the main text, so we explain it in Appendix A.

Finally we can obtain more relations by using the fact that the Picard-Fuchs operators $\mathcal{L}_{l,i}$ annihilate the maximal cuts,

$$
\begin{aligned}
\partial_{z_i}\underline{\Pi}(\underline{z})\Sigma_l\partial_{\underline{z}}^{\underline{k}}\underline{\Pi}(\underline{z}) &= \partial_{z_i}C_{\underline{k}}(\underline{z}) - \underline{\Pi}(\underline{z})\Sigma_l\partial_{\underline{z}}^{\underline{k}+1_i}\underline{\Pi}(\underline{z}) \\
&= \partial_{z_i}C_{\underline{k}}(\underline{z}) + \underline{\Pi}(\underline{z})\Sigma_l\sum_{\underline{p}/(\underline{k}+1_i)}^{|p|=l-1}\frac{\beta_{i,\underline{p}}^{(l)}(\underline{z})}{\beta_{i,\underline{k}+1_i}^{(l)}(\underline{z})}\partial_{\underline{z}}^{\underline{p}}\underline{\Pi}(\underline{z})\,.
\end{aligned}
\tag{89}
$$

We can keep taking derivatives of eq. 82 and using the Picard-Fuchs operator to gain more and more relations. Eventually we will have enough relations to construct a generic-mass version of the matrix

$$
\mathbf{Z}_l(\underline{z}) = \begin{pmatrix}
\underline{\Pi}(\underline{z})^T\Sigma_l\underline{\Pi}(\underline{z}) & \dots & \underline{\Pi}(\underline{z})^T\Sigma_l\partial_{\underline{z}}^{s_\lambda}\underline{\Pi}(\underline{z}) \\
\vdots & \ddots & \vdots \\
\partial_{\underline{z}}^{s_\lambda}\underline{\Pi}(\underline{z})^T\Sigma_l\underline{\Pi}(\underline{z}) & \dots & \partial_{\underline{z}}^{s_\lambda}\underline{\Pi}(\underline{z})^T\Sigma_l\partial_{\underline{z}}^{s_\lambda}\underline{\Pi}(\underline{z})
\end{pmatrix}\,.
\tag{90}
$$

Note that the differentials that go into $\mathbf{Z}_l(\underline{z})$ are the same as the ones that are used to construct the Wronskian. The matrix has the same symmetry as in the equal-mass case, to wit: $\mathbf{Z}_l(\underline{z}) = (-1)^{l+1}\mathbf{Z}_l(\underline{z})^T$. One can show that in the generic-mass case the inverse Wronskian is still expressible as

$$
\mathbf{W}(\underline{z})^{-1} = \Sigma_l\mathbf{W}(\underline{z})^T\mathbf{Z}(\underline{z})^{-1}\,.
\tag{91}
$$

In the generic-mass case, we are then once again able to describe the inverse Wronskian linearly in terms of entries in $\mathbf{W}_l(\underline{z})$.

## 4.2 Unipotent differential equation

We proceed, following essentially the same analysis as in the equal-mass case. First we need to find the form of the inhomogeneous term in eq. 30. We may once again act with the Picard-Fuchs operators on our basis of master integrals $\underline{J}_l(\underline{z})$. Doing so, we find that only the very first element is trivial. The non-trivial elements are complicated polynomials in all scales. To concentrate on the relevant structure, we will here simply express the inhomogeneous part as

$$
\mathbf{N}_l(\underline{z}) = \begin{pmatrix}
0 \\
\sum_i^{l+1}S_{1,i}(\underline{z})dz_i \\
\vdots \\
\sum_i^{l+1}S_{l-1,i}(\underline{z})dz_i
\end{pmatrix}\,,
\tag{92}
$$

where the $S_{i,j}$'s are determined by the inhomogeneities obtained from eq. 32.

We now define the ratios $\tau_k = \varpi_k(\underline{z})/\varpi_0(\underline{z})$, which we will later take as unipotent periods in our basis. We then use eq. 31 to express the differential equation our master integrals satisfy, in the $L$-basis. As opposed to the equal-mass case, the entries in the inverse Wronksian are not necessarily proportional to just one element (c.f. eq. 50), but can in general be linear combinations of all elements of the Frobenius basis. This is for instance the case at $l = 3$. With this in mind we can write the differential equation eq. 31 as

$$
dL_k(\underline{z}) = \sum_{i=0}^{l-1}c_{k,i}\big(X_i(\underline{z})\tau_i(\underline{z}) + D_i(\underline{z})\big)\,,
\tag{93}
$$

where $c_{k,i} \in \{-1, 0, 1\}$ and,

$$X_k(\underline{z}) = \alpha_{k,\underline{0}}(\underline{z})\varpi_0(\underline{z}) + \sum_{\underline{s} \in T} \alpha_{k,\underline{s}}(\underline{z})\partial_{\underline{z}}^{\underline{s}}\varpi_0(\underline{z}),$$

$$D_k(\underline{z}) = \sum_{\underline{s} \in T} \sum_{\substack{\underline{m},\underline{s} \\ \underline{n} \neq \underline{0} \\ \underline{m}+\underline{n}=\underline{s}}} \alpha_{k,\underline{s}}(\underline{z})\partial_{\underline{z}}^{\underline{m}}\varpi_0 \partial_{\underline{z}}^{\underline{n}}\tau_k(\underline{z}).$$

(94)

The set $T$ contains all vectors such that for $\underline{s} \in T$ the derivative $\partial_{\underline{z}}^{\underline{s}}J_{l,0}(\underline{z})$ is a master integral. The $\alpha$ coefficients are one-forms of the form $\alpha(\underline{z}) = \sum_{i}^{l+1} \omega_i(\underline{z})dz_i$, where the $\omega_i$ are rational functions.

We now define a new basis, which will allow us to build a unipotent differential equation, by expanding the $L_k(\underline{z})$ basis with all $\tau_k(\underline{z})$ elements:

$$T_l(\underline{z}) = \left(L_{l,1}(\underline{z}), ..., L_{l,\lambda}(\underline{z}), \tau_1(\underline{z}), ..., \tau_\lambda(\underline{z}), 1\right).$$

(95)

This basis satisfies the unipotent differential equation

$$d\underline{T}(\underline{z}) = \mathbf{N}_l(\underline{z})\underline{T}_l(\underline{z}).$$

(96)

The nilpotent matricies for $l = 3, 4$ have the following structure:

$$\mathbf{N}_3(\underline{z}) = \begin{pmatrix} 0 & \overset{\times 5}{\cdots} & 0 & 0 & 0 & 0 & X_5 & D_5 \\ 0 & \cdots & 0 & -X_2 & -X_3 & -X_4 & 0 & -D_2 - D_3 - D_4 \\ 0 & \cdots & -X_1 & 0 & -X_3 & -X_4 & 0 & -D_1 - D_3 - D_4 \\ 0 & \cdots & -X_1 & -X_2 & 0 & -X_4 & 0 & -D_1 - D_2 - D_4 \\ 0 & \cdots & -X_1 & -X_2 & -X_3 & 0 & 0 & -D_1 - D_2 - D_3 \\ 0 & \cdots & 0 & 0 & 0 & 0 & 0 & X_0 + D_0 \\ 0 & \cdots & 0 & 0 & 0 & 0 & 0 & d\tau_1 \\ 0 & \cdots & 0 & 0 & 0 & 0 & 0 & \vdots \\ 0 & \cdots & 0 & 0 & 0 & 0 & 0 & d\tau_5 \\ 0 & \cdots & 0 & 0 & 0 & 0 & 0 & 0 \end{pmatrix},$$

(97)

$$\mathbf{N}_4(\underline{z}) = \begin{pmatrix} 0 & \overset{\times 11}{\cdots} & 0 & 0 & 0 & 0 & 0 & 0 & 0 & 0 & 0 & 0 & X_{11} & D_{11} \\ 0 & \cdots & 0 & 0 & 0 & 0 & 0 & -X_6 & 0 & 0 & 0 & 0 & 0 & -D_6 \\ 0 & \cdots & 0 & 0 & 0 & 0 & 0 & 0 & -X_7 & 0 & 0 & 0 & 0 & -D_7 \\ 0 & \cdots & 0 & 0 & 0 & 0 & 0 & 0 & 0 & -X_8 & 0 & 0 & 0 & -D_8 \\ 0 & \cdots & 0 & 0 & 0 & 0 & 0 & 0 & 0 & 0 & -X_9 & 0 & 0 & -D_9 \\ 0 & \cdots & 0 & 0 & 0 & 0 & 0 & 0 & 0 & 0 & 0 & -X_{10} & 0 & -D_{10} \\ 0 & \cdots & X_1 & 0 & 0 & 0 & 0 & 0 & 0 & 0 & 0 & 0 & 0 & D_1 \\ 0 & \cdots & 0 & X_2 & 0 & 0 & 0 & 0 & 0 & 0 & 0 & 0 & 0 & D_2 \\ 0 & \cdots & 0 & 0 & X_3 & 0 & 0 & 0 & 0 & 0 & 0 & 0 & 0 & D_3 \\ 0 & \cdots & 0 & 0 & 0 & X_4 & 0 & 0 & 0 & 0 & 0 & 0 & 0 & D_4 \\ 0 & \cdots & 0 & 0 & 0 & 0 & X_5 & 0 & 0 & 0 & 0 & 0 & 0 & D_5 \\ 0 & \cdots & 0 & 0 & 0 & 0 & 0 & 0 & 0 & 0 & 0 & 0 & 0 & -X_0 - D_0 \\ 0 & \cdots & 0 & 0 & 0 & 0 & 0 & 0 & 0 & 0 & 0 & 0 & 0 & d\tau_1 \\ 0 & \cdots & 0 & 0 & 0 & 0 & 0 & 0 & 0 & 0 & 0 & 0 & 0 & \vdots \\ 0 & \cdots & 0 & 0 & 0 & 0 & 0 & 0 & 0 & 0 & 0 & 0 & 0 & d\tau_{12} \\ 0 & \cdots & 0 & 0 & 0 & 0 & 0 & 0 & 0 & 0 & 0 & 0 & 0 & 0 \end{pmatrix},$$

(98)

where we have suppressed the explicit $\underline{z}$ dependence.

We note that the nilpotent matrices are very similar in structure to the intersection matrices for these cases. In this sense, it is relatively simple to construct the nilpotent matrix once one knows the intersection matrix.

There is an additional constraint that the differential equation must hold in order to define a well-defined symbol, namely an integrability condition of the form,

$$d\mathbf{N}_l = \mathbf{N}_l \wedge \mathbf{N}_l. \tag{99}$$

This condition is trivial in the equal-mass case, as that case involves only a single variable. Here, it is nontrivial. We have checked this condition using a series expansion of the matrix entries for small $\underline{z}$, and it holds, with $d\mathbf{N}_l$ and $\mathbf{N}_l \wedge \mathbf{N}_l$ separately nonzero.[7]

## 4.3 Symbol

We are now ready to construct a symbol on our master integrals. We once again define a unipotent matrix of the form in eq. 62 using the nilpotent matrices found in the previous section. Similarly to the equal-mass case the symbols are all of length two. We list all relevant results for the three- and four-loop cases.

$l = 3$:

$$
\begin{aligned}
\mathcal{S}([\xi_i, \xi_i]) &= 1, \\
\mathcal{S}([1, L_1]) &= [D_5] + [d\tau_5|X_5], \\
\mathcal{S}([1, L_2]) &= -[d\tau_2|X_2] - [d\tau_3|X_3] - [d\tau_4|X_4] - [D_2] - [D_3] - [D_4], \\
\mathcal{S}([1, L_3]) &= -[d\tau_1|X_1] - [d\tau_3|X_3] - [d\tau_4|X_4] - [D_1] - [D_3] - [D_4], \\
\mathcal{S}([1, L_4]) &= -[d\tau_1|X_1] - [d\tau_2|X_2] - [d\tau_4|X_4] - [D_1] - [D_2] - [D_4], \\
\mathcal{S}([1, L_5]) &= -[d\tau_1|X_1] - [d\tau_2|X_2] - [d\tau_3|X_3] - [D_1] - [D_2] - [D_3], \\
\mathcal{S}([1, L_6]) &= [X_0] + [D_0], \\
\mathcal{S}([1, \tau_k]) &= [d\tau_k].
\end{aligned}
\tag{100}
$$

$l = 4$:

$$
\begin{aligned}
\mathcal{S}([\xi_i, \xi_i]) &= 1, \\
\mathcal{S}([1, L_1]) &= [D_{11}] + [d\tau_{11}|X_1 1], \\
\mathcal{S}([1, L_2]) &= -[d\tau_6|X_6] - [D_6], \\
\mathcal{S}([1, L_3]) &= -[d\tau_7|X_7] - [D_7], \\
\mathcal{S}([1, L_4]) &= -[d\tau_8|X_8] - [D_8], \\
\mathcal{S}([1, L_5]) &= -[d\tau_9|X_9] - [D_9], \\
\mathcal{S}([1, L_6]) &= -[d\tau_{10}|X_{10}] - [D_{10}], \\
\mathcal{S}([1, L_7]) &= [d\tau_1|X_1] + [D_1], \\
\mathcal{S}([1, L_8]) &= [d\tau_2|X_2] + [D_2], \\
\mathcal{S}([1, L_9]) &= [d\tau_3|X_3] + [D_3], \\
\mathcal{S}([1, L_{10}]) &= [d\tau_4|X_4] + [D_4], \\
\mathcal{S}([1, L_{11}]) &= [d\tau_5|X_5] + [D_5], \\
\mathcal{S}([1, L_{12}]) &= -[X_0] - [D_0], \\
\mathcal{S}([1, \tau_k]) &= [d\tau_k].
\end{aligned}
\tag{101}
$$

---

[7] We thank Stefan Weinzierl for bringing this to our attention.

### 4.4 Coaction

Now that we have the symbols defined, we can calculate the coaction on our master integrals as discussed for the equal-mass case. We find the following:

$l = 3$:

$$
\begin{aligned}
\Delta(L_{3,1}(\underline{z})) =& L_{3,1}(\underline{z}) \otimes 1 + 1 \otimes D_5 + 1 \otimes [d\tau_5|X_5] + \tau_4 \otimes X_5, \\
\Delta(L_{3,2}(\underline{z})) =& L_{3,2}(\underline{z}) \otimes 1 - 1 \otimes [d\tau_2|X_2] - 1 \otimes [d\tau_3|X_3] - 1 \otimes [d\tau_4|X_4] - 1 \otimes (D_2 + D_3 + D_4) \\
& - \tau_2 \otimes X_2 - \tau_3 \otimes X_3 - \tau_4 \otimes X_4, \\
\Delta(L_{3,3}(\underline{z})) =& L_{3,3}(\underline{z}) \otimes 1 - 1 \otimes [d\tau_1|X_1] - 1 \otimes [d\tau_3|X_3] - 1 \otimes [d\tau_4|X_4] - 1 \otimes (D_1 + D_3 + D_4) \\
& - \tau_1 \otimes X_1 - \tau_3 \otimes X_3 - \tau_4 \otimes X_4, \\
\Delta(L_{3,4}(\underline{z})) =& L_{3,4}(\underline{z}) \otimes 1 - 1 \otimes [d\tau_1|X_1] - 1 \otimes [d\tau_2|X_2] - 1 \otimes [d\tau_4|X_3] - 1 \otimes (D_1 + D_2 + D_4) \\
& - \tau_1 \otimes X_1 - \tau_2 \otimes X_2 - \tau_4 \otimes X_4, \\
\Delta(L_{3,5}(\underline{z})) =& L_{3,5}(\underline{z}) \otimes 1 - 1 \otimes [d\tau_1|X_1] - 1 \otimes [d\tau_2|X_2] - 1 \otimes [d\tau_3|X_3] - 1 \otimes (D_1 + D_2 + D_3) \\
& - \tau_1 \otimes X_1 - \tau_2 \otimes X_2 - \tau_3 \otimes X_3, \\
\Delta(L_{3,6}(\underline{z})) =& L_{3,6}(\underline{z}) \otimes 1 + 1 \otimes X_0 + 1 \otimes D_0, \\
\Delta(\tau_k) =& \tau_k \otimes 1 + 1 \otimes [d\tau_k].
\end{aligned}
\tag{102}
$$

$l = 4$:

$$
\begin{aligned}
\Delta(L_{4,1}(\underline{z})) =& L_{4,1}(\underline{z}) \otimes 1 + 1 \otimes [d\tau_{11}|X_{11]} + 1 \otimes D_{11} + \tau_{11} \otimes X_{11} \\
\Delta(L_{4,2}(\underline{z})) =& L_{4,2}(\underline{z}) \otimes 1 - 1 \otimes [d\tau_6|X_1] - 1 \otimes D_1 - \tau_6 \otimes X_1, \\
\Delta(L_{4,3}(\underline{z})) =& L_{4,3}(\underline{z}) \otimes 1 - 1 \otimes [d\tau_7|X_2] - 1 \otimes D_2 - \tau_7 \otimes X_2, \\
\Delta(L_{4,4}(\underline{z})) =& L_{4,4}(\underline{z}) \otimes 1 - 1 \otimes [d\tau_8|X_3] - 1 \otimes D_3 - \tau_8 \otimes X_3, \\
\Delta(L_{4,5}(\underline{z})) =& L_{4,5}(\underline{z}) \otimes 1 - 1 \otimes [d\tau_9|X_4] - 1 \otimes D_4 - \tau_9 \otimes X_4, \\
\Delta(L_{4,6}(\underline{z})) =& L_{4,6}(\underline{z}) \otimes 1 - 1 \otimes [d\tau_{10}|X_5] - 1 \otimes D_5 - \tau_{10} \otimes X_5, \\
\Delta(L_{4,7}(\underline{z})) =& L_{4,7}(\underline{z}) \otimes 1 + 1 \otimes [d\tau_1|X_6] + 1 \otimes D_6 + \tau_1 \otimes X_6, \\
\Delta(L_{4,8}(\underline{z})) =& L_{4,8}(\underline{z}) \otimes 1 + 1 \otimes [d\tau_2|X_7] + 1 \otimes D_7 + \tau_2 \otimes X_7, \\
\Delta(L_{4,9}(\underline{z})) =& L_{4,9}(\underline{z}) \otimes 1 + 1 \otimes [d\tau_3|X_8] + 1 \otimes D_8 + \tau_3 \otimes X_8, \\
\Delta(L_{4,10}(\underline{z})) =& L_{4,10}(\underline{z}) \otimes 1 + 1 \otimes [d\tau_4|X_9] + 1 \otimes D_9 + \tau_4 \otimes X_9, \\
\Delta(L_{4,11}(\underline{z})) =& L_{4,11}(\underline{z}) \otimes 1 + 1 \otimes [d\tau_5|X_{10}] + 1 \otimes D_{10} + \tau_5 \otimes X_{10}, \\
\Delta(L_{4,12}(\underline{z})) =& L_{4,12}(\underline{z}) \otimes 1 - 1 \otimes X_0 - 1 \otimes D_0, \\
\Delta(\tau_k) =& \tau_k \otimes 1 + 1 \otimes [d\tau_k].
\end{aligned}
\tag{103}
$$

Unlike in the $l = 3$ equal-mass case, here there can be no direct comparison with existing expressions. We only remark that the structure we observe appears to be quite consistent across loop orders, up to the complexity of the coefficients $X_k$ and $D_k$. It is again possible to transform back to the original $\underline{J}_l(\underline{z})$ basis, with the transformation $\underline{J}_l(\underline{z}) = \mathbf{W}_l(\underline{z})\underline{L}_l(\underline{z})$. Doing so we again find that for $l \geq 3$ the symbol length is three.

## 5 Conclusions

We have constructed a coaction and symbol for the sunrise integral at arbitrary loop order, both for the equal-mass case and for the generic-mass case. We built these coactions and symbols by re-arranging the differential equations satisfied by the sunrise master integrals into a unipotent form, augmenting them with ratios of maximal cuts $\tau_i$. We find that, counter to

our experience in the case of polylogarithms, the lengths of these symbols do not increase with loop order: instead, they saturate at length three to all loops. This suggests that our symbol entries are increasing in complexity with loop order, which may mean they satisfy even more complicated relations along the lines of the symbol-prime of ref. [86]. At two and three loops, our symbols and coactions should in principle be comparable with those derived for the equal-mass sunrise at two loops in ref. [62], for the equal-mass sunrise at three loops in ref. [68], and for the unequal-mass sunrise at two loops in ref. [86]. While the structures we observe are in general similar, showing direct equivalence would involve developing a formalism to convert between symbols and coactions involving different choices of bases of unipotent and semi-simple periods, a task which we reserve for future work.

We highlight two choices in our construction, which if varied could yield substantially different results. First, we made a choice to expand our basis with the ratios $\tau_i$, and not with additional functions. In general, one could imagine introducing more intermediate functions related by differential equations, leading to longer symbols. Such functions would not be Feynman integrals, at least not in the same dimension as the sunrise integrals we consider, but this should not be especially unfamiliar: when constructing polylogarithmic functions to bootstrap scattering amplitudes as in refs. [26–38], one must in general consider integrals that are not themselves Feynman integrals. An example of such a choice that appeared after our initial preprint leverages the notion of Calabi-Yau operators to further decompose the equal-mass sunrise and ice cream cone integrals [83–85]. It would be interesting to see if these notions can be generalized to the generic-mass case and to other multivariate problems more generally.

Second, we made a choice in our separation of our integrals into semi-simple and unipotent factors. Different choices can lead to different lengths, and to substantially different symbols more generally, by factoring out different transcendental functions from the functions considered by the symbol. While we believe our choice is reasonable (motivated in part by the symbol and coaction on elliptic multiple polylogarithms), it is not the only possible choice, and a different choice may be more informative in other circumstances.[8]

It is important to note that the letters of our symbol contain higher powers of logarithms around the MUM point. As such, taking a limit that reduces the sunrise to a polylogarithm should have a nontrivial effect on our symbol, almost certainly not directly yielding a polylogarithmic symbol. In future work we would like to investigate the approach to pseudothresholds, the most straightforward such limit to study.

In general, we expect a symbol for Calabi-Yau integrals that matches the length of polylogarithmic symbols to require a somewhat different construction. It would need to involve a substantially larger basis, likely as part of a basis of functions more general than the sunrise master integrals alone. It would be extremely interesting to construct such a basis, which may demand investigation of more general diagrams involving Calabi-Yau manifolds along the lines of [88–91].

# Acknowledgements

We thank Kilian Bönisch, Matthias Wilhelm, and Cristian Vergu for helpful discussions. MvH also thanks the Mainz Institute for Theoretical Physics for hospitality while this paper was being prepared for submission, and useful comments from the attendees of the workshop Elliptic Integrals in Fundamental Physics.

---

[8]For an example of a formalism that handles this aspect differently, we refer the reader to ref. [87] which appeared after this work appeared on the arXiv, and which applies the original formalism of ref. [63] to the two-loop sunrise.

**Funding information** This work was supported by the Danish National Research Foundation (DNRF91), the research grant 00015369 from Villum Fonden, and a Starting Grant (No. 757978) from the European Research Council.

# A Solution to differential equations satisfied by generic-mass Yukawa-couplings

Solving the differential equations in eq. 88 can be done in an iterative fashion. We will start by considering the explicit example of solving the equations for $l = 3$. We will in the end comment on how the $l = 3$ case generalizes quite easily for higher loop orders.

In the case of $l = 3$, the Griffiths Transversality condition (eq. 33) only has non-trivial solutions for $|k| = 2$. We start with the explicit example of $\underline{k} = (2, 0, 0)$. In this case, we have three equations (one for each $i$):

$$\partial_{z_1} C_{2,0,0}(\underline{z}) = 2 \frac{\beta^{(l)}_{l,(2,0,0)}(\underline{z})}{\beta^{(l)}_{l,(3,0,0)}(\underline{z})} C_{2,0,0}(\underline{z}), \tag{A.1}$$

$$\partial_{z_1} C_{1,1,0}(\underline{z}) = \frac{1}{2} \partial_{z_2} C_{2,0,0}(\underline{z}) + \frac{\beta^{(l)}_{l,2,0,0}(\underline{z})}{\beta^{(l)}_{l,(2,1,0)}(\underline{z})} C_{2,0,0}(\underline{z}), \tag{A.2}$$

$$\partial_{z_1} C_{1,0,1}(\underline{z}) = \frac{1}{2} \partial_{z_3} C_{2,0,0}(\underline{z}) + \frac{\beta^{(l)}_{l,(2,0,0)}(\underline{z})}{\beta^{(l)}_{l,(2,0,1)}(\underline{z})} C_{2,0,0}(\underline{z}). \tag{A.3}$$

The first of these equations, namely eq. A.1, is a differential equation only in the function $C_{2,0,0}(\underline{z})$. This differential equation has the general solution

$$C_{2,0,0}(\underline{z}) = c_1(z_2, z_3) \exp\left(2 \int_1^{z_1} dt \frac{\beta^{(l)}_{l,k}(t, z_2, z_3)}{\beta^{(l)}_{l,(3,0,0)}(t, z_2, z_3)}\right), \tag{A.4}$$

where $c(z_2, z_3)$ is an integration constant. Note that due to the symmetry of $z_1, z_2, z_3$ in eq. 88, the solutions for $C_{0,2,0}$ and $C_{0,0,2}$ are the same as in eq. A.4, only with the role of $z_1$ exchanged with $z_2$ and $z_3$ respectively. The solution for $C_{2,0,0}, C_{0,2,0}$ and $C_{0,0,2}$ can be used to solve the differential equations involving $C_{1,1,0}, C_{1,0,1}, C_{0,1,1}$ by simple integration.

This procedure generalizes to higher loop order. One starts at $\underline{k}_1 = (0, ..., l - 1, ..., 0)$, to find the solutions of the type in eq. A.4. One then moves on to $\underline{k}_2 = (0, ..., l - 2, , 0, ..., 1, ..., 0)$ and solves the equations using the functions $C_{\underline{k}_1}$ just obtained. It is possible to continue in this fashion until one reaches the differential equations corresponding to $\underline{k}_l = (1, 1, ..., 1, 0, 1, ..., 1)$. Due to the way the differentials are present in eq. 6, we note that the $\mathbf{Z}_l(\underline{z})$ will only depend on Yukawa couplings of the type $C_{1,...,1,0,1,...,1}(\underline{z})$.

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
