# Peer review of "A Symbol and Coaction for Higher-Loop Sunrise Integrals"

_SciPost Physics Core, doi:SciPost Phys. Core 6, 050 (2023)_

## Round 3 · Referee Report · Anonymous (Referee 1) · 2023-4-29

Strengths

1- the paper deals with a very timely topic and which has important consequences both on formal questions on scattering amplitudes and feynman integrals, and on practical methods to evaluate them

2- the paper is very well written and is very clear

Weaknesses

1- the paper proposes a definition of a symbol and coaction map for l-loop sunrise graphs, but it does not go beyond the simple definition to show how that definition can be useful in practice to answer general questions on scattering amplitudes.

Report

The paper introduces a definition of a symbol map and coaction for the sunrise graphs at l loops. The paper is well written and the exposition is very clear. The formatting is largely good and clear, except a small issue that I identified, see requested changes below.
On the unsatisfactory side, as the authors admit themselves, their definition is rather arbitrary, inasmuch as it depends on the separation in semi-simple and unipotent factors adopted. The most interesting observation from this point of view is indeed how the length of the symbol defined, depends strongly from this definition, to the point that with their definition, complexity does not increase from loop to loop in terms of length of the symbol, which remains constant. This implies that it is the entry of the symbol, and therefore the relations among them, that become increasingly complicated.

I do not believe this is a reason not to publish this paper on scipost, since the topic is interesting. Nevertheless, what is missing, in my opinion, is a motivation of why this definition is useful at all. In the polylogarithmic world, the symbol and coaction operations allow us not only to "dissect" the analytic properties of the answer and to use this information to find relations among apparently different expressions, to perform analytic continuation, to bootstrap complicated amplitudes etc. This is mainly due to the fact that the symbol dissects polylogs into differential of logarithms, the relations among which are all well understood, at least in principle.

In this sense, it looks to me that the competing definition (admittedly valid only for the elliptic case) given by Wilhelm and Zhang in Ref[81] seems to be going more in the right direction of what a “symbol map should achieve”. Of course, in the present case, the authors are aiming to studying more general structures, beyond elliptic, and one cannot expect to understand everything at once. Nevertheless, I think the authors should consider elucidating better why their symbol map is useful, possibly with some explicit example: what information can they extract from it and how can they imagine to use it to perform or simplify a calculation?

Requested changes

1- check the typesetting after eq 36, since the formula in the text goes outside of the margins

2- I think the authors should define what is an intersection matrix before or right after introducing it in eq 37, as this is not part of the normal tool kit of a high energy physicists that might be reading this paper

3- the authors should clearly state that the comments after eq.50 referring to Griffits transversality and the fact that, due to this, the inverse Wronskian depends linearly on the periods, are not new in this paper but where worked out in details in ref [74].

4- After eq 56 there is a strange reference to eq (3.2) which does not seem to exist in the text.

5- as explained in the report, I think the issue with the length of the symbol discussed by the authors is one of the interesting ones and should be discussed more in detail. In particular, can the authors clarify why they believe that their definition, where the length does not change with the loops, should be more useful than another definition?

6- as a final comment, the authors refer in various points to the fact that the maximal cuts solve the homogeneous differential equations for a set of feynman integrals (first occasion is I believe before eq 28). I think they should refer to where these statements have been made first: it was first realised in hep-ph/0406160 that the physical discontinuity of the 2 loop sunrise solves the homogeneous equation and that this could be used to find an analytic expression for the period of the corresponding elliptic curve. This observation was (to my knowledge) not obvious for a long time and was only generalised to the max cut and to the leading singularities of any Feynman integrals in a series of papers: 1610.08397, 1701.07356, 1704.04255 , 1704.05465, 1705.03478

---

## Round 3 · Referee Report · Anonymous (Referee 2) · 2023-5-9

Strengths

1- This paper generalises the notion of symbol and coaction beyond elliptic Feynman integrals. The idea is interesing for the Feynman integral and/or scattering amplitude community, especially for those interested in integrals beyond elliptic curves. 2- This paper is well written and organised.

Weaknesses

1- Many typos 2- Lack of concrete examples for usage

Report

This paper constructs a symbol and coaction for $l$-loop Banana integrals for equal-mass and unequal-mass cases. On top of ref [74], the authors enlarge the basis with ratios of periods around the MUM point. Surprisingly, by the specific choice of this basis, symbols saturate at length three to all loops, which is the main result of this paper.

This paper is well organized but has many typos, which I would require the authors to correct in the following. After the corrections, I recommend publishing this paper on Scipost.

On top of that, I suggest the authors provide more comparisons with existing literature, at least for the equal-mass case at three loops, i.e., 2109.15251 and 2207.12893. This will definitely reduce the criticism about lacking concrete examples for usage. In this context, it may be more transparent to see the advantage of the authors' choice for decomposition into semi-simple and unipotent parts.

A side remark from my opinion about the author's choice: the ratios introduced in the enlarged basis can be related to so-called $Y$-invariants for Calabi-Yau operators, which have been applied for equal-mass Bananas and ice-cone integrals: 2212.08908 and 2212.09550, which appeared after the preprint.

Requested changes

1- Label the range for $k$ in (9) 2- I suggest that the authors can saying something about what are $\underline{\Pi}$ and the interaction matrix $\Sigma_l$ around (33). A few sentences are enough. This will be helpful for non-experts or readers that are not familiar with ref [74]. Furthermore, please distinguish the intersection matrix with the Frobenius solutions $\Sigma_{l,k}$ by stating the difference or change the notation. 3- Please beautify the typeset of the formula in the paragraph between (36) and (37) 4- In (38), it should be $\partial_z^{l-2}$ 5- There misses $T$ for $\underline{\Pi}$ in (48) 6- Underline for $N_l$ in (52) 7- $l\to k$ in (53) 8- Replace eq. (3.2) with the right hyperlink
9- In (60), the row index $i$ and column $j$ are exchanged due the (58) and (59). The authors may think how to make it more rigorous. 10- Below (65), I believe it should be $\mathcal{S}\left(\left[\tau_{1}, L_{2,2}(z)\right]\right)$ 11- In (66), missing $dz$ 12- In (81), $\partial_{z}^{\underline{s}}\to \partial_{\underline{z}}^{\underline{s}}$ 13- Below (81), $1,\leq i \leq l+1\to 1\leq i \leq l+1$ 14- (84) use the $n$ as summation index, but it was stated as fixed at the beginning 15- (84) misses underling for $z$ in the end 16- (88) $k_n\to\underline{k}_n$ 17- (90) the transpose symbol is missing

---

## Round 4 · Referee Report · Anonymous (Referee 2) · 2023-5-30

Report

The comments and requirement for changes have been implemented.

---

## Round 4 · Referee Report · Anonymous (Referee 1) · 2023-6-1

Report

All points in my report have been addressed, my opinion is that the paper can be published in present form on Scipost.

---

## Round 4 · Author Response

As recommended by the editor and referees, we resubmit with the minor changes requested.

---

## Round 4 · List of Changes

We have added additional discussion regarding the relationship of this notion of the symbol to others, in particular regarding its length, clarifying in particular which aspects relate to a choice of basis of functions and which to the division between semi-simple and unipotent functions. This discussion appears in the Conclusions and at the end of section 3, and references the subsequent work mentioned by the second referee. We believe that this expanded discussion should address both referees' concerns regarding motivation of the formalism and comparison to the existing literature and related formalisms, including point (5) of the second referee.

We have addressed the typos mentioned in the second referee report as (1, 3, 4, 5, 6, 7, 8, 9, 10, 11, 12, 13, 14, 15, 16, 17), two of which were also mentioned in the first referee report as (1, 4). In addition we have corrected several cases of missing underlines on vector quantities.

We have addressed comment (2) of the first referee and comment (2) of the second referee by adding a definition of the intersection matrix after eq. (33). As recommended by the second referee, we have added a footnote on page 9 to mitigate confusion of notation.

We have addressed comment (3) of the first referee by clarifying the origin of the points made at the end of section 3.1.

We have included the references suggested in comment (6) of the first referee in a footnote on page 7.

---

## Editorial Decision

published